# Co-Creation of a School-Based Motor Competence and Mental Health Intervention: Move Well, Feel Good

**DOI:** 10.3390/children10081403

**Published:** 2023-08-17

**Authors:** Lauren Clifford, Richard Tyler, Zoe Knowles, Emma Ashworth, Lynne Boddy, Lawrence Foweather, Stuart J. Fairclough

**Affiliations:** 1Movement Behaviours, Health, Wellbeing, and Nutrition Research Group, Department of Sport and Physical Activity, Edge Hill University, St. Helens Road, Ormskirk L39 4QP, UK; tylerr@edgehill.ac.uk (R.T.); faircls@edgehill.ac.uk (S.J.F.); 2The Physical Activity Exchange, Research Institute for Sport and Exercise Sciences, Liverpool John Moores University, 5 Primrose Hill, Liverpool L3 2EX, UK; z.r.knowles@ljmu.ac.uk (Z.K.); l.m.boddy@ljmu.ac.uk (L.B.); l.foweather@ljmu.ac.uk (L.F.); 3School of Psychology, Liverpool John Moores University, Liverpool L3 5UX, UK; e.l.ashworth@ljmu.ac.uk

**Keywords:** co-creation, intervention development, motor competence, mental health, children, fundamental movement skills, physical activity

## Abstract

Low motor competence (MC) and inhibited psychosocial development are associated with mental health difficulties. Improving children’s MC through school-based physical activity interventions emphasising psychosocial development may therefore be a mechanism for promoting positive mental health. This study describes and provides reflective insights into the co-creation of ‘Move Well Feel Good’, a primary school physical activity intervention to improve children’s MC and mental health. Class teachers, school leaders, physical activity specialists, and children (aged 8–9 years) participated in a series of co-creation workshops. Stakeholders’ knowledge and experiences were integrated with existing research evidence using creative methods (e.g., post-it note tasks, worksheets, and drawings) to facilitate discussion. The co-creation process culminated in stakeholder consensus voting for one of three proposed intervention ideas. Children cited physical and mental health benefits, enjoyment with friends, and high perceived competence as motives for being physically active. Opportunities to develop MC across the different segments of the school day were identified by adult stakeholders, who perceived children’s lack of resilience, an overloaded curriculum, and poor parental support for physical activity as barriers to intervention implementation. The chosen intervention idea received six out of a possible twelve votes. Co-creation projects are specific to the contexts in which they are implemented. This study reinforces the complex nature of school-based intervention development and highlights the value of engaging with stakeholders in co-creation processes.

## 1. Introduction

Mental health difficulties amongst children are a leading cause of health-related issues worldwide [1]. Globally, 25.2% and 20.5% of children and young people suffer from depression and anxiety, respectively [2], with the prevalence of at-risk children in England increasing from one in nine in 2017 to one in six in 2022 [3]. Mental health difficulties in childhood are associated with a range of negative outcomes, such as low academic achievement, decreased quality of life, impaired physical health, illegal substance use, and poor social relationships [4,5,6], and are a strong predictor of poor mental health in adulthood [7]. Mental health difficulties across the life course are strongly associated with childhood socioeconomic disadvantage [8]. Further, low socioeconomic status (SES) is associated with a high number of stressors for families related to finances, social relations, employment situations, and health complaints, in comparison to higher SES families [9,10]. As a result, children from low SES families are two to three times more likely to experience poor mental health, with stronger associations reported in children under 12 years of age [11]. Moreover, there is evidence that lockdown restrictions during the COVID-19 pandemic negatively affected children’s mental health [12], with families from lower SES backgrounds disproportionately affected [13,14]. Therefore, interventions to enhance mental health and wellbeing among children, particularly those experiencing socioeconomic disadvantage, are warranted and timely.

The COVID-19 lockdown restrictions also resulted in increases in children’s digital screen use and decreases in physical activity (PA), particularly for structured activities such as school PE lessons and sport participation [15,16], which are essential for the development of motor competence (MC) [17]. MC relates to the development and performance of human movement, representing an individual’s ability to perform skilfully on a wide range of motor tasks, including fundamental movement skills (FMS) [17,18]. Positive reciprocal relationships exist between PA and MC, whereby children with high MC are more likely to participate in PA than children with low MC, and children that engage in higher levels of PA typically have higher levels of MC [18,19,20,21]. Therefore, reduced PA during the COVID-19 pandemic is likely to be reflected by attenuation of MC or slowing of MC development, particularly among children from low SES families who have fewer opportunities for structured and high-quality PA [22]. Poor MC is also associated with inhibited psychosocial development (e.g., self-esteem, resilience, and perceived competence) and internalising mental health difficulties (e.g., anxiety and depression), with these associations potentially being mediated by limited social support mechanisms [23,24] and low self-perceptions [25,26]. Improving children’s MC may therefore be a mechanism for improving children’s mental health through enhancing aspects of psychosocial development.

These inter-relationships between children’s MC, mental health, and their mediators are described in the Elaborated Environmental Stress Hypothesis (EESH), which posits that poor MC predisposes children to internalising mental health difficulties via interactions with environmental stressors such as low self-esteem, low social support, physical inactivity, and being overweight. These stressors can in turn be ‘buffered’ by social and personal resources such as peer and parental support and perceived competence [23]. Butler et al. [27] highlighted the particular importance of peer support as a protective factor among children with poor mental health, whereby strong peer support had the equivalent protective impact of family and school adult support combined. When considered in the context of schools, these findings demonstrate the crucial context of this setting in fostering peer relationships to promote mental health and resilience for children with and without supportive home environments [27].

School-based programmes can reach many children irrespective of family backgrounds, and thus may be a strong influence on children’s health and development [28]. It is further acknowledged that mental health education is now expected to be delivered in the school curriculum in England, demonstrating the necessity and importance of promoting positive mental health amongst children and young people [29]. Primary school MC interventions can be efficacious for improving motor skills [30,31], and there is some limited evidence that they can also enhance mental health and wellbeing [32,33]. However, it has been recognised that school-based intervention programmes that are solely based on academic theories may lack the contextual understanding of individual school needs [34,35]. Furthermore, Lendrum and Humphrey [36] highlight the importance of social validity (i.e., the perceived acceptability, feasibility, and utility) in school-based interventions, and its relationship to intervention outcomes. They conclude that interventions should not only achieve their theoretical outcomes, but also be feasible, acceptable, and effectively implemented in real-world settings [36]. Co-creation has recently emerged as an approach to remedy this, by providing opportunities for key stakeholders to participate in intervention development processes, where they would previously be traditionally excluded from decision-making [37]. The shared stakeholder ownership of the process can provide a context-sensitive basis for acceptable interventions, potentially increasing the likelihood of their feasibility and of them being implemented as intended, and thus effective [38]. However, until now evidence of the processes of co-created interventions are only just emerging. To address this gap, the aim of this study was to describe and provide reflective stakeholder and researcher insights into the co-creation process used to create the Move Well Feel Good (MWFG) programme, which is a primary school intervention to improve children’s mental health through development of MC and psychosocial skills. This novel study was guided by the EESH and employed a design-thinking approach to intervention co-creation, defined as collaborative intervention development by academics working alongside other stakeholders with relevant experiential knowledge [39].

## 2. Materials and Methods

### 2.1. School and Stakeholder Recruitment

The MWFG primary school programme was co-created with school stakeholders in the West Lancashire region of northwest England between March and July 2022. The study received ethical approval from the Edge Hill University Science Research Ethics Committee (ETH2122-0062). Eligibility criteria for participating primary schools were (1) being situated in school postcode-linked English Indices of Multiple Deprivation deciles 1–3 (i.e., situated in high deprivation areas) [40], and (2) having >18% free school meal eligibility. An email invitation to participate was sent to Head Teachers of nine primary schools that met the eligibility criteria. Six schools indicated their interest and were recruited. One withdrew before the study commenced due to Head Teacher illness. School sizes varied between 207–264 children (mean = 233) and were all one form entry. As an incentive to take part, each of the five participating schools received a bag of PA equipment equivalent to the value of £300. A designated school leader (School Head Teacher or member of School Leadership Team) and Year 4 class teacher from each school were selected to represent their school in a series of co-creation workshops. Two physical education (PE) specialists from West Lancashire Sport Partnership (WLSP; an organisation that provides PE, PA, and school sport teaching and support to West Lancashire schools) were also recruited to participate in the co-creation process. All school and WLSP adult participants provided written informed consent before the study commenced. To aid child participant recruitment, short, animated videos were created detailing the project information, and the measures that were going to be collected during the co-creation phase and the subsequent intervention feasibility phase of the project. The animated videos were presented to Year 4 children (aged 8–9 years) in school and were sent out via online school communications for parents/carers to access and view (see Appendix A for an example). As an incentive to take part, the children were each offered a £10 Amazon voucher to be received at the end of the study. Parental written informed consent and child assent was obtained for all child participants before the study commenced.

### 2.2. Intervention Development

The co-creation research involved children, class teachers, school leaders, and PE specialists. Intervention co-creation was conducted through a 6-stage process facilitated by the research team. The stages aligned with the Double Diamond Design Approach (DDDA) by employing divergent and convergent thinking processes. These reflected how stakeholders discovered, defined, developed, and delivered solutions to the ‘problem’ of how to improve children’s MC and mental health, and how best to facilitate real-world implementation in school contexts [41]. The DDDA is a well-regarded approach within design thinking and has been used in other school-based co-creation research [42].

The co-creation process was planned into four time-efficient workshops (between 60–180 min) to reduce the participation burden on stakeholders (Table 1). Adult stakeholders completed a variety of tasks in single and multiple stakeholder groups to reach a consensus on the components and content of a school-based MC and mental health intervention. When participants worked in single stakeholder groups (SSG), they were grouped based on their main role, for instance, all class teachers together. When the participants were mixed into multi-stakeholder groups (MSG), each group consisted of at least one class teacher, one school leader, and one PE specialist. The adult stakeholders were working towards producing stakeholder generated ideas for the MWFG programme, which culminated in a consensus majority vote. The programme idea that received the most votes would be delivered in the new school term. To engage the children in the process without them feeling inhibited in expressing their thoughts in the presence of the adults that regularly teach them, they operated in an SSG in stages 1 and 2, after which their discussion points were shared in stages 3 and 4 with the adult stakeholder groups. Data collection tools included digital audio recorders, observation notes, participant worksheets, and follow-up interviews, which were used collectively to capture the events of the co-creation workshops.

**Table 1 children-10-01403-t001:** Summary of the co-creation workshops used to create the MWFG programme mapped onto the DDDA stages.

Workshop and Duration	Participant Group	Stakeholder Group	Workshop Tasks	DDDA Stage	Thinking Processes
1 (1 h)	Children	SSG	Aim: To check knowledge and understanding of psychosocial skills whilst generating an enthusiasm to be involved in the project and building rapport with the children. • A 1-h practical MC session, based on activities used in the Dragon Challenge dynamic MC assessment [43]. • Following researcher instructions and peer-demonstrations, children performed the individual skills (i.e., jumping, throwing, catching, dribbling), rotating around three working areas for locomotor, object control, and stability movement skills. • Activity prompt cards which included technical points were used by children for self- and peer-feedback (Appendix A). • Preliminary activity performed to check children’s understanding of psychosocial skills affecting PA participation (e.g., social skills, social support, self-esteem, and resilience) [23]. Phrases such as “High fiving or clapping for a classmate” and “Not giving up” were presented to the children where they matched the phrases to the different psychosocial skills. • Concluded with a question-and-answer plenary task to reinforce the importance of psychosocial skills when in PA and sport contexts, and in classroom and home settings.	1	Convergent
2 (1 h)	Children	SSG	Aim: To get the children to develop their ideas for what a movement skill intervention might involve. • Interactive discussion groups with eight children from each school (*n* = 40). • Children discussed and recorded the types of physical activities that they enjoyed doing and why. • Children were provided with an A3 worksheet and coloured post-it notes and were encouraged to use different coloured post-it notes to represent the types of physical activities and the reasons for enjoying them. • Children repeated the task but for physical activities that they did not enjoy doing. • Children were asked to think about what their ideal PA programme would look like. To assist them, a worksheet with key headings of ‘who’, ‘what’, ‘when’, ‘where’, and ‘how’ based on the TiDier checklist was provided to reflect key intervention design considerations [44]. • Collated responses were subsequently shared in the adult stakeholder workshops.	1	Divergent
1 (1 h)	Adults	SSG	Aim: To provide stakeholders with the required baseline knowledge and understanding on children’s MC and psychosocial skills, to engage in the co-creation process and subsequent workshops. • A 1-h pre-recorded online presentation that provided key information about the underlying concepts for the project, including definitions of key terms, the different types of FMS that contribute to MC, and the importance of developing these for lifelong PA. • Tasks were integrated throughout with an online knowledge quiz at the end to check understanding of the content and to monitor engagement. • Stakeholders were able to engage with this at a time that was convenient for them up to two weeks before workshop 2.	1	N/A
2 (2 h)	Adults	SSG and MSG	Aim: To identify how MC and psychosocial skills could be better promoted and enhanced throughout the school day. • Stakeholders worked in SSG to identify what opportunities were already in place for children to become proficient in FMS and develop positive wellbeing, influenced by the psychosocial environment in various segments of the school day (e.g., classroom lessons, PE, break and lunch time, and before and after school). • Stakeholders then worked in MSG to discuss and report how FMS and psychosocial skills could be better promoted in the different segments of the school day. • Stakeholders identified the barriers they faced in helping children to develop movement skills and psychosocial competences in the school environment. • Nine from an anticipated twelve stakeholders attended.	2	Convergent and Divergent
3 (3 h)	Adults	MSG	Aim: To draft and develop programme ideas for improving MC and mental health in the school context. • Eight from an anticipated twelve stakeholders attended, thus, two, rather than the planned three groups of four were formed. • As an icebreaker task, the two groups engaged in a discussion of children’s PA preferences and how well these aligned to the adults’ perceptions of them. • The current evidence for effective FMS and psychosocial development interventions (delivery context, delivery personnel, pedagogical considerations, intervention dose) was presented to inform the stakeholders’ ideas (Appendix A). • Using the children’s ideas, the research evidence, and their knowledge and understanding of FMS and psychosocial skills from prior workshops, the two groups were asked to develop their ideas for the MWFG programme that would be feasible and acceptable in a typical primary school setting. • The programme had to be conducted within the school environment, within the extended school day (8 a.m.–5 p.m.), over a maximum of 12 weeks, with a minimum delivery dose of one formal activity per week. • Both groups presented their ideas back to the other group at various intervals to discuss and gain feedback before continuing to refine their ideas.	3, 4 and 5	Convergent and Divergent
4 (1 h)	Adults	Individual	Aim: Adult stakeholders to vote on the programme ideas created in workshop 3 and reach a consensus on the preferred programme. • Programme ideas from workshop 3 were presented back to the stakeholder groups via a pre-recorded online presentation. • Three programme ideas were presented, one from each group developed during workshop 3, plus a third option which was generated by the research team using a combination of ideas from both groups, in addition to some of the research evidence. • Each stakeholder had one vote to cast for their preferred option. • The results of the consensus vote with details of next steps were communicated to the stakeholders via email.	6	Convergent

Note: N/A—Not applicable as thinking processes were not applied.

### 2.3. Semi-Structured Interviews

Semi-structured interviews were conducted with selected teacher workshop participants to provide insight into the stakeholders’ perspectives of the co-creation process. Teachers were recruited based on their availability and interest to participate. A semi-structured interview guide was developed to ensure key topic areas were covered, while allowing for unanticipated responses. The interview questions were developed by the research team following their observations of the workshops. These questions were reflective of the stakeholders’ feedback, engagement, and enjoyment of the workshops (see interview guide in Appendix A). The interviews were transcribed verbatim. As this paper purposely aims to describe the co-creation process of the MWFG programme, no formal analysis was undertaken. Instead, verbatim quotes from the interviews are presented with the intention to illuminate discussion points and allow the participants voices to be represented.

## 3. Results

Observations and quantitative findings from the co-creation process are presented below and in the order in which the workshop phases were conducted.

### 3.1. Child and Adult Workshop 1

In each school, all children in Year 4 participated in workshop 1. Class sizes varied between 24 and 31 (mean = 28) children across the five schools. Twelve adult stakeholders received the online presentation, and six completed the knowledge quiz prior to attending workshop 2.

### 3.2. Child Workshop 2

The children’s responses to the questions that they were asked regarding their PA preferences for inclusion in a new programme are presented in Figure 1.

Across the five schools, the children reported a variety of physical activities that they enjoyed participating in, including team sports, leisure activities, and active play. The most prevalent activities that the children reported enjoying were football (*n* = 17), running (*n* = 13), biking (*n* = 13), swimming (*n* = 11), basketball (*n* = 10), and tag (*n* = 8). The children’s motives for being physically active were physical and mental health benefits (“…because it (running) gets rid of energy. It helps you get to sleep easier” (Boy—School 5)), fun and enjoyment (“…because I like dribbling and shooting, it’s really fun for me (basketball)” (Girl—School 3)), learning new skills (“To learn new skills” (Girl—School 1]), high perceived competence (“football because I am a great defender” (Boy—School 1)), and social elements such as being with friends (“I enjoy ‘stuck in the mud’ because people work together” (Girl—School 3)). Elements of competition and winning were highlighted by a minority of children as a motive for being physically active (“I like running because it is very fun, and you can compete” (Boy—School 4)), although this was not common across all groups, whereby having fun was usually deemed more important (“I like racing because it’s not about winning, it’s about having fun” (Girl—School 2)).

The activities that children most frequently reported disliking were rugby (*n* = 10), ballet (*n* = 9), gymnastics (*n* = 7), football (*n* = 6), basketball (*n* = 6), and dancing (*n* = 3). Low perceived competence was the most prominent reason why children reported not enjoying an activity (“I don’t like swimming because I sink like a rock” (Boy—School 4)), alongside having a lack of understanding of the activity (“I do not enjoy rugby because I don’t know how to play it” (Girl—School 3)). An overview of the children’s PA preferences, reasons why, and examples of the completed worksheets is presented in Appendix A.

### 3.3. Adult Workshop 2

The adult stakeholders reported an extensive number of opportunities for children to develop FMS and psychosocial skills across the school day (Table 2). The opportunities related to the promotion of physically active learning, brain breaks via digital educational technologies such as ‘Wake Up Shake Up’, a greater balance between inclusion and competition in PE, and having structured timetables to keep PA consistent across play times. The data that were collected during this task are extracted, presented, and further contextualised in Table 2 relative to each stakeholder group’s perspectives, and an example of the activity resources is presented in Figure 2.

Subsequent group discussions highlighted the consequences of the COVID-19 pandemic on not only the children’s PA levels, but also their ability to interact socially and resolve conflict, causing a challenge for schools when reintroducing activity and play during lunch and play times. One school leader commented.

“Children’s ability to deal socially with fall outs, arguments, challenge, you know we’ve talked about resilience and stuff so one of the things that were talking about is…we’re having to put more and more structured supervised play on during lunch times and play times because the children can’t play in a way that they used to”.(School Leader–Male—School 2)

It was apparent that these psychosocial challenges upon returning to school after the COVID-19 pandemic were central to the barriers that the adult stakeholder groups reported (Table 3). These barriers related to the overloaded curriculum, lack of parent/carer support, low degree of child resilience, social influences, lack of equipment, lack of teacher knowledge regarding cross-curricular PA delivery, and various factors external to the school setting. Each stakeholder group discussed the barrier that they believed to be the most important to overcome. The class teachers identified the most prominent barrier as children’s reactions to either lack of success or celebrating others’ success. One teacher stated, 

“… they either see themselves as not being able to achieve as well as the known superstar which puts them off, or just an inability to be proud of somebody else and want to strive for that themselves”.(Class teacher—Female—School 3)

School leaders believed a lack of parent/carer support and understanding of the importance of PE and PA was the most prominent barrier (“We wrote ‘support’ but then we also put ‘understanding’, it’s not just a lack of support, it’s a lack of understanding, and the children are going to be affected by that” (School Leader—Male—School 2)). The PE specialists reported the most important barrier to be social influences of parents/carers and others (“The children see it as not being cool, but they see it as not being cool because their parents have got that negative ‘I’m not interested’ attitude” (PE Specialist—Female—School 5)). This was manifested in children not having PE uniform for lessons, being self-conscious about their bodies, and not being interested in being active.

### 3.4. Adult Workshop 3

The main task in workshop 3 was for the adult stakeholders to work together in their groups to create and develop their ideas for the MWFG programme, incorporating FMS and psychosocial skills. Prior to developing intervention ideas, feasibility and acceptability factors deemed essential for a prospective intervention programme were discussed. These are summarised in Table 4 and broadly relate to school leadership, training, curriculum integration, resources, and timetabling.

Two MSG were formed to draft and develop ideas for the MWFG programme. An example of one of the worksheets produced for this task is illustrated in Figure 3. Group one’s ideas were abstract and somewhat under-developed with limited emphasis on FMS. The group focused on activities that could be applied across the curriculum, particularly those that de-emphasised competition and competence comparisons, such as forest school, hide-and-seek, treasure hunts, and circuit sessions. It was envisioned that these activities would be used as a vehicle to improve FMS and psychosocial skills amongst children, whilst providing a greater balance between inclusion and competition to address the identified lack of resilience. In comparison, group two’s idea had greater structure (Figure 3) but also lacked detail and an emphasis on how FMS would be applied. The group highlighted that their structure would span multiple subjects across a 12-week period that could be repeated across terms. It was envisioned by the group that this structure would be repeated using different skills and activities depending on the FMS and PE focus that term. Their motivation for this structure was to embed the programme into school culture, as opposed to being a stand-alone programme. Because the groups’ ideas were abstract with a limited focus on FMS, the research team were unable to translate them into a deliverable programme. Further, the research evidence presented to the stakeholders was generally not applied, and the research team felt that this would limit the likelihood of intervention feasibility and subsequent effectiveness. Therefore, the research team generated a third programme idea for stakeholders to consider. The ‘hybrid’ programme combined ideas from both groups and included elements reflecting the research evidence presented in workshop 3, to ensure that the instruction and teaching of FMS was embedded alongside pedagogical principles. Due to the time of year (summer term), it was not possible to extend workshop 3 or to schedule a further workshop to engage with the stakeholders to obtain their input regarding the hybrid idea 3 before the end of term. The hybrid idea 3 was communicated to stakeholders in the online pre-recorded presentation in workshop 4.

### 3.5. Consensus Vote

Of the twelve stakeholders involved in the co-creation process, eleven engaged with the pre-recorded online presentation and the consensus vote. Group one’s idea received 1 vote, group two’s idea received 4 votes, and the hybrid idea 3, which had some researcher input, received 6 votes. Therefore, idea 3 was confirmed for feasibility implementation in participating schools in the following Autumn term. The components of the consensus MWFG programme are displayed in Table 5.

## 4. Discussion

This study aimed to describe and provide reflective insights into the co-creation process used to create the MWFG programme, and to discuss the issues that this process generated. Several key learnings emerged relating to (1) stakeholders’ engagement and attendance, (2) stakeholders’ knowledge and competence, (3) time constraints, and (4) conflicting visions and motives. These are discussed below and, based on the lessons learned from our experiences, suggested recommendations for researchers planning to follow a similar process are presented at the end of each section.

### 4.1. Stakeholder Engagement and Attendance

A lack of available time and the competing demands for teachers’ time accounted for observed engagement and attendance issues [45]. Of the twelve stakeholders who were expected to attend workshop 2 and 3, only eight attended both. Despite the consistent number of stakeholders at both workshops, different teachers from the same schools attended the different workshops, which caused a lack of staff continuity. Inconsistent stakeholder attendance has been reported as a challenge to co-creation research, particularly when collaborating with multiple stakeholder groups [46]. The inclusion of multiple stakeholder groups provided a significant challenge to the scheduling of the workshops, especially amongst school staff who had busy schedules and limited available time for anything outside of the normal curriculum [45,47]. The competing demands of staff time have been reported to affect the scheduling of face-to-face collaboration in previous co-creation research, which limits stakeholders’ participation in the process [45,48,49]. The competing priorities for teachers’ time meant that some stakeholders were unable to attend. In the instances where stakeholders were unable to attend workshops, no alternate members of staff were sent as replacements, which meant some schools were underrepresented. Likewise, some teachers had to leave the workshops early because they were called back to school to deal with unexpected pupil-related situations. Stakeholders’ reflections of the process highlighted the challenging reality of collaborating with primary schools, particularly in the summer term (June and July).

“I don’t think it’s ever going to be easy and you’re never going to nail the perfect way of working with primary schools in times like the summer term, because you’re always going to run into those pressure points where primary schools are being pulled in different directions to do different things at that point in the year”.(School Leader—Male)

Stakeholder discussions highlighted how the timing of the co-creation process presented its own challenges for teachers’ engagement and attendance. Due to inflexible project timelines and the time restrictions surrounding the research process [50], the co-creation workshops were scheduled in the summer term so the co-created programme could be implemented in the following Autumn term (September–December) of the new school year. Despite scheduling the co-creation workshops around the stakeholders’ availability [49], the competing demands for teachers’ time was further heightened in the summer term. Staff were under pressure to meet various deadlines before the end of term and were perceived to be burnt out [51], and this tension was felt across all the stakeholders involved in the process (“It was a busy time of the year wasn’t it I think for everyone at that time, especially headteachers and teachers” (PE Specialist—Male)). Subsequent stakeholder discussions, however, highlighted that although the summer term provided challenges to engagement in the current study, every term provides its own difficulties.

“It’s hard because if you want anything to roll out successfully from the autumn term you have to do it in the summer term but trying to get people to engage in the summer term…there’s never a good time, there’s never a good time”.(School Leader—Male)

It is clear that there is never an optimal time to collaborate with primary schools, and every school term will bring about its own challenges, particularly in participatory research which can be time-consuming for teachers. The competing priorities for teachers’ time, coupled with the lack of available time that teachers have due to pressures of the ever-expanding curriculum, added adversity to the engagement and attendance issues that we observed [45,48]. Researchers wanting to conduct a similar process with schools should be aware of the challenges associated with the competing demands of teachers’ job roles and their lack of available time and account for these issues in the planning of project timelines in collaboration with teacher stakeholders [52]. Suggesting specific one-size-fits -all timeframes is not appropriate, however; researchers should consider investing time at the start of the process to fully engage with teachers in advance of the co-creation activities beginning to understand their roles, commitments, and expectations. This can serve several purposes, such as building and maintaining relationships, anticipating pressure points for time demands (e.g., end of term and student assessments), developing appropriate timelines, and obtaining a shared commitment from school leaders and designated teacher stakeholders to attend and engage with co-creation activities (e.g., having a named alternate person if attendance is not possible and being cognitively present during workshops) [52,53].

### 4.2. Stakeholders’ Knowledge and Competence

It was clear that the stakeholder groups had differing capabilities regarding their knowledge and understanding of FMS. It is well documented that generalist teachers have a lack of knowledge and understanding of PE principles, particularly in primary schools where PE is commonly delivered by generalist teachers without a PE specialism [54,55]. This lack of knowledge stems from an absence of specialist training for PE [56], where generalist teachers only receive basic training (i.e., minimum of 6 h) via their teacher education [57,58]. We used the DDDA to employ convergent and divergent thinking processes in the workshop activities [41]. This allowed the stakeholders to work collaboratively to answer the same questions at every stage of the co-creation process [41]. On reflection, dividing the stakeholders up to work on different tasks based on their expertise could have been more advantageous to programme development. For example, PE specialists were more knowledgeable and had a greater understanding of the teaching and delivery of FMS [59,60], whereas the class teachers had greater knowledge of the pupils’ educational and pastoral needs [61]. As all the participating schools had PE specialists deliver their PE curricula, an alternative workshop approach would have been for the PE specialists to focus on the FMS element of programme development, and the school leaders and class teachers to focus on the psychosocial skills element. Optimising the expertise of PE specialists and generalist teachers in this way may have positively contributed to the level of detail provided in the groups’ final programme ideas, whilst also reducing the likelihood of stakeholders producing somewhat abstract and unformed ideas. Although the stakeholder groups were varied throughout the workshops (e.g., SSG and MSG), the grouping of stakeholders could have been further optimised to maximise the knowledge and expertise of the participating stakeholders [53]. Upon reflection of this, a recommendation for the DDDA is that what is suitable for one person or group might be unsuitable for another, particularly in co-creation projects where there are multiple stakeholders with differing knowledge and expertise [41]. Therefore, it is important to have flexibility within groupings to choose suitable stakeholders for specific activities to not limit innovation and creativity [41]. Furthermore, restructuring the workshop activities to optimise the integration between PE specialists and school staff could have been a better use of the limited time available with stakeholders in the face-to-face workshops. Researchers should consider the skills and knowledge of the stakeholders and reflect on how best to harness their specific expertise to achieve the aims of each workshop or co-creation activity.

### 4.3. Time Constraints

It is widely recognised that co-creation is demanding of time, resources, and commitment for researchers and stakeholders involved in the process [62,63]. Despite our best efforts to plan and account for this, interactions with stakeholders took up more time than was anticipated [64]. For example, the duration of workshop 3 was three hours, but this was not long enough for the stakeholders to create and draft their programme ideas. In the workshop, the stakeholders were asked to create programme ideas that incorporated FMS and psychosocial skills, and that were informed by the children’s ideas, their work from workshop 2, and the research evidence. On reflection, the expectation that the stakeholders could digest and synthesise this volume of information and create a programme idea by the end of the three-hour workshop was unrealistic [65].

Discussions with stakeholders highlighted how the time constraints in workshop 3 and the subsequent input from the research team led to feelings of nonaccomplishment and a lack of ownership over the process. The stakeholders would have preferred to have more time to collaborate with the other stakeholders to develop a combined programme idea, rather than the research team generating the combined hybrid version. Optimising the integration between the stakeholder groups within the three-hour workshop would have been beneficial to not risk de-valuing the co-creation ethos. However, given the time constraints experienced in workshop 3, it was not possible to do this, especially as the teachers needed to return to school afterwards. Further, extending the co-creation process was not feasible, as it was essential that the MWFG programme was designed and consensus was received before the end of the school year, to ensure that programme resources could be developed in time for delivery. The challenges of this type of research have been echoed by others. For example, O’Brien and Dadswell [66] highlighted how the disruptions caused by the school holidays not only contributed to the limited time to collect data with stakeholders, but also to the limited time to receive consensus before the end of the school year. Further, the disparity in time frames in co-creation research is acknowledged to present its own challenges, which are exacerbated by the short timelines commonly associated with funded research projects [67]. Inflexible project timelines coupled with limited teacher availability in the summer term meant that it was not possible to extend co-creation in the current study [50]. Furthermore, communicating with stakeholders and organising the co-creation workshops took a considerable amount of time, and was not without its difficulties given the teachers’ lack of time and competing priorities [67]. Working with stakeholders in advance to gain context-specific understanding of the time and resources needed would have enhanced the quality and outcomes of the co-creation process [53].

### 4.4. Conflicting Visions and Motives

Conflicting priorities in participatory research are common, particularly amongst multidisciplinary groups where there are contrasting views and expertise involved in the research process [46,64]. Throughout the co-creation process, it was apparent that there were conflicting visions and motives amongst the teacher stakeholders. For example, when creating programme ideas teachers were heavily focused on their own views and perceptions and how a programme would work best in their school, as opposed to thinking and working holistically in a collaborative way. 

“Every school will have their own difficulties, accommodations that need to be made… it was more us, me thinking how can we make this work for us?”.(Class Teacher/PE Specialist—Female)

Ponsford et al. [48] highlighted similar challenges, particularly the difficulty of ensuring co-created interventions were locally appropriate, while maintaining the opportunity to use existing theory and build on evidence-based approaches. Furthermore, they discussed that it was not always appropriate or possible to implement the advice of the stakeholders where their views contradicted existing best practice and the logic of interventions [48]. We observed how the stakeholders did not integrate the research evidence into their thinking and planning of the MWFG programme. This caused a conflict between researchers and stakeholders, particularly as teachers viewed the research evidence solely as an academic agenda rather than as something to align their ideas with [46]. Integrating the research evidence with reality required compromise from stakeholders and the research team, which did not always guarantee resolution for all stakeholders [49].

These difficulties were associated with an imbalance of power in decision making between the stakeholders and the research team [53,62]. Tensions surrounding power dynamics in school-based participatory research have previously been identified [48] and have been described as ‘a balancing act between two social worlds’ [68]. Barke et al. [69] reported similar challenges due to researchers and stakeholders holding different ideas of what a project’s aim and remit was. It is possible that the teacher stakeholders lacked a clear understanding of the co-creation process and its aims. 

“There seemed to be on that day two different trains of thoughts that I know that ended up with you producing was it three models, three proposals that we all voted on. And I don’t know whether that was an intention at the time, or just a product of that day at Edge Hill, that this was the only kind of way we could find to map our way forward”.(School Leader—Male)

During the initial headteacher meetings, the research team clearly outlined the process of co-creation and its aims, including the consensus voting and that the ‘winning’ idea would be implemented in each school. However, it cannot be assumed that the communications were accurately cascaded to the school leaders and class teachers participating in the co-creation process, particularly when only one headteacher engaged in the whole process. Had there not been time constraints surrounding co-creation in the current study, engaging with the participating stakeholders prior to the workshops would have been beneficial to clarify the co-creation process and its aims, overcome any early conflicts, and manage stakeholder expectations [70]. This approach has been advocated by others through initiation of the co-creation process with a ‘needs analysis’ to identify and resolve challenges at an early stage [46,49] to ensure stakeholders feel that they are equal contributors to the process of project design and delivery from the beginning [67].

## 5. Conclusions

The MWFG programme was developed following a co-creation process involving children, school leaders, class teachers, and PE specialists. We have described and reflected on each stage of the process and the complexities that were raised whilst reporting honestly, offering perspectives from the research as it happened, proposing alternatives and creating key considerations for researchers which related to stakeholders’ engagement, attendance, knowledge and competence, time constraints, and conflicting visions and motives for taking part. Clearly, co-creation projects such as this are specific to the contexts in which they are implemented and the perspectives of the participants. For this reason, a ‘one size fits all’ approach is not applicable or appropriate. However, there may be certain characteristics of school-based co-creation projects that share some commonalities, which we reflect in our recommendations for other researchers. These relate to investing sufficient time at the start of the process to engage with stakeholders to clarify the co-creation process and its aims, gaining context-specific understanding of the time and resources needed, understanding and managing stakeholders’ roles, commitments, and expectations (including those of the research team), and overcoming any early conflicts. Engaging with stakeholders at all stages of co-creation projects in an open and clear way is crucial to shared success.

## Figures and Tables

**Figure 1 children-10-01403-f001:**
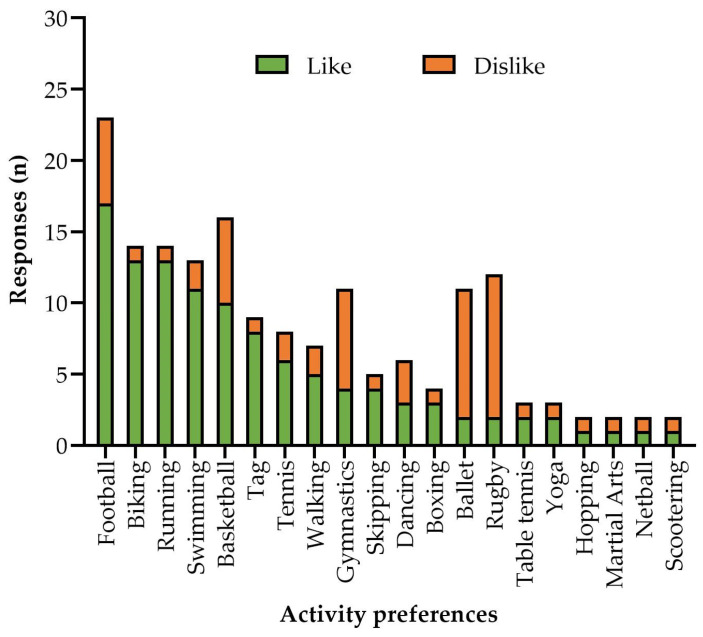
Summary of the activities that children reported to like and dislike in workshop 2.

**Figure 2 children-10-01403-f002:**
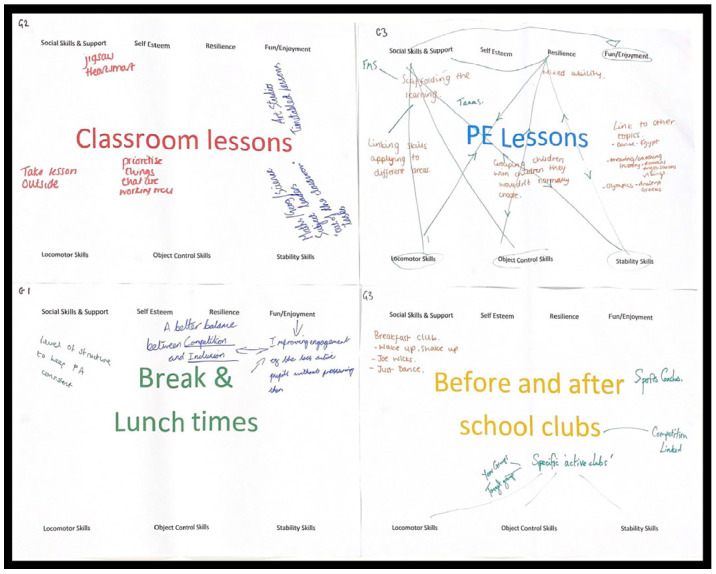
Examples of the work produced by the adult stakeholders in workshop 2. Note. Red pens represent class teachers, blue pens represent school leaders, and green pens represent PE specialists.

**Figure 3 children-10-01403-f003:**
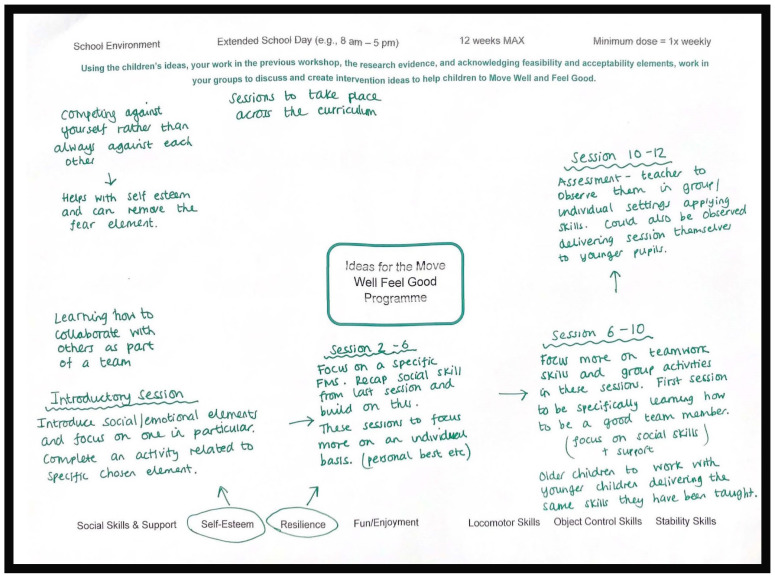
A snapshot of the work produced by adult stakeholders in workshop 3.

**Table 2 children-10-01403-t002:** Opportunities identified by the adult stakeholders to promote MC and psychosocial skills across the school day.

	Class Time	PE Lessons	Break and Lunch Times	Before and after School
Class Teachers	- Taking the lesson outside - Jigsaw and ‘Heartsmart’ [Digital educational technologies] - Prioritising things that are working well - Promotion of active learning, i.e., Active maths/literacy - Outdoor provision - Location change - Movement of places	- Sports Day - Swimming - Extra yoga sessions - Grouping children with children they would not normally choose. - Scaffolding the learning linking and applying FMS to different areas - Mixed ability - Links to other cross-curricular topics [LINKED TO FMS and PS] [e.g., dance—Egypt, Olympics—Ancient Greeks, throwing and catching–history]	- Buddy system games [Linked to self-esteem and resilience] - Structured games led by lunchtime staff - Wake up, shake up in the hall or class [Digital technology for short bursts of PA]	- Continue sports before and after school, i.e., Yoga - Gardening club run by grandparents - Art sessions in studio–timetabled lessons and community - Breakfast club -Wake up, shake up [Digital technology for short bursts of PA] - Joe Wicks [Online PA resource] - Just dance [Virtual dance game series]
School leaders	- Brain Gym/Wake ‘n’ shake [Digital technology for short bursts of PA] - More opportunities for movement - Maths/Geography/Science subject leaders–“out of the classroom tasks” [Physically active lessons] - Art Studio timetabled lessons	- Yoga—6-week programme [sustain the programme after it finishes]	- Improving engagement of the less active pupils without pressuring them [Better balance between competition and inclusion linked to fun/enjoyment] - Key stage 2 mentors [role models for key stage 1 and early years foundation stage children] - Year 6 buddy systems - Timetable structure for varied sports and activities - Gardening club - Forest School area (lunch club)	- Better training for breakfast club and after school club staff to improve activity and provision. - Structured breakfast club activities.
PE specialists	- Brain breaks across all classes. [It has to be consistent, i.e., across the whole school]	- Balance between inclusion and making it competitive/challenge [Linked to resilience, fun, and enjoyment]. - More corporation between PE specialists and school staff [Linked to locomotor, object control, and stability skills]	- Level of structure to keep PA consistent [Having a routine structure for PA at break and lunch times] - Playground leaders - Sports coach involvement - Rota for facilities and equipment	- Sports coaches - Specific ‘active clubs’ for year groups and target groups [Linked to locomotor, object control, and stability skills]. - Competition links

Note. FMS—Fundamental Movement Skills; PS—Psychosocial skills; PA—Physical Activity; Self-defining in [square brackets] for context.

**Table 3 children-10-01403-t003:** Barriers reported by the stakeholder groups.

Overarching Barrier	Specific Examples	Class Teachers	School Leaders	PE Specialists
Overloaded curriculum	- Timetable pressures- Demands of the school curriculum on school staff- Lesson lengths (flexibility is needed)	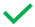	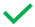	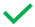
Lack of parent/carer support	- Parents/carers lack of understanding of the importance of PE and PA- Overbearing parent/carer worries- Lack of parent/carer consent- Lack of interest in PE and PA from parents/carers which impacts children’s PA (PA perceived as ‘uncool’)- Pressures of work and finance (parents/carers)	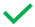	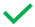	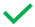
Lack of resilience from children	- Children not willing to improve once ‘failure’ occurs- Children’s reactions to either lack of success or celebrating others- Lack of resilience and enthusiasm from pupils- Poor behaviour- Low energy (no breakfast)	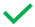	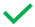	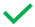
Children’s social influences	- Self-conscious (social media, body image)- Peer groups- Junk food- Role models and mainstream media			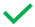
Equipment and facilities	- Limited space in the school grounds- Lack of equipment- PE kit is not brought into school or is incorrect for the activity			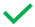
Lack of knowledge and training	- Lack of knowledge and ideas for creative delivery- Fear and nerves	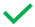		
External factors	- Budget- Flexibility- Time- Fatigue from late nights	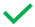		

**Table 4 children-10-01403-t004:** Feasibility and acceptability elements reported by stakeholder groups.

Feasibility and Acceptability Factors
Senior leadership team driving the programme among staff and children and getting staff members on board.
Staff meetings, teaching, and training alongside PE specialists.
Cross curricular links to allow teachers to incorporate an active lesson once weekly (i.e., active maths and literacy).
Embedding the programme into PE lessons in school.
A fully resourced programme with ‘ready to go’ planning and resources that can be used despite any staff changes and as part of continuous professional development.
Easily timetabled.

**Table 5 children-10-01403-t005:** Components of the MWFG programme.

FMS instruction and teaching via PE	- 1 dedicated lesson/week. - FMS instruction, differentiated practice, application. - Pre-determined combinations of locomotor, object/control, and stability skills would be focused on in teaching blocks to allow consistent ‘mirroring’ reinforcement of these skills elsewhere in the curriculum (e.g., classroom lessons). - Specific psychosocial concepts (social skills and social support) integrated into the PE lessons in teaching blocks aligned to the FMS focus. - Opportunities for self- and peer-teaching of various movement skills.
Reinforcement of psychosocial concepts and opportunities to practice FMS embedded across the curriculum	- Natural environment (e.g., Forest school activities) - PE environment (e.g., application to different PE activities) - Classroom and school grounds as appropriate (e.g., physically active learning)
Supplementary practice opportunities	Self-directed individual or group ‘Skill Snacks’ that are short, fun, and allow children to practice the skills being taught in each block during PE: - At break times facilitated by equipment and resource task cards (play focus) - At home facilitated by resource task cards and online resources (e.g., video clips) and proactively promoted via schools’ communication/social media. - Potential to link into goal setting to encourage personal improvement.
Culminating event	- Celebration of the learning over the course of the programme. - Peer-teaching of FMS to younger pupils.
Additional information	- Duration of programme = 9 weeks (10 weeks in total including half-term) - Delivery mainly by PE specialists. - Supported by class teachers and break-time supervisors, where applicable. - Training and resources provided by EHU research team.
Pedagogy	- SAAFE Framework principles (supportive, active, autonomous, fair, enjoyable) - FMS Assessment for Learning (task-oriented, self- and peer-assessment)

Note. FMS: Fundamental movement skill; EHU: Edge Hill University.

## Data Availability

The data are not publicly available due to participant consent not being obtained for this.

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
