# Peer review of "Co-Creation of a School-Based Motor Competence and Mental Health Intervention: Move Well, Feel Good"

_children, 2023, doi:10.3390/children10081403_

Round 1

Reviewer 1 Report

Review of children-2527555

This manuscript is a nice contribution for the study of children, school, physicomotoric study, etc. The study is well planned and executed. It is recommended for publication after some minor problems are revised, as follows:

  1. Because this manuscript discusses about child-friendly cities for children aged 8-9, which are in their phases of primary education, then additional references about child-friendly schools are highly suggested to be added also in this manuscript.
  • Journal of Asian Architecture and Building Engineering 21 (2022) 865-883 https://doi.org/10.1080/13467581.2021.1928506

  1. Please add list of abbreviations (before the list of references)

  1. Figure 1 and line 208-213: Based on Figure 1, there are several likes and dislikes of physical activities. The activities with high dislikes are basketball, gymnastics, ballet and rugby. One of the reason for a student dislikes rugby is shown in line 212-213. How about basketball, ballet and gymnastics?

  1. Line 214: Figure S3 --> letter e was forgotten.
  2. Reference 29: 26th
  3. Reference 57: 28th

Author Response

Please see t

Comment #

Comment

Response

1

This manuscript is a nice contribution for the study of children, school, physicomotoric study, etc. The study is well planned and executed. It is recommended for publication after some minor problems are revised, as follows:

Thank you for these positive comments.

2

Because this manuscript discusses about child-friendly cities for children aged 8-9, which are in their phases of primary education, then additional references about child-friendly schools are highly suggested to be added also in this manuscript.

Journal of Asian Architecture and Building Engineering 21 (2022) 865-883 https://doi.org/10.1080/13467581.2021.1928506

We respectfully disagree with the reviewer’s interpretation of the manuscript being about or discussing child-friendly cities for children. We are also unclear what is meant by ‘child-friendly’ schools, because by definition school environments facilitate children’s learning and development. Further, the article suggested by the reviewer was not relevant to our study. For these reasons we have not included the additional references suggested.

3

Please add list of abbreviations (before the list of references)

A list of abbreviations has been added at the end of the manuscript.

4

Figure 1 and line 208-213: Based on Figure 1, there are several likes and dislikes of physical activities. The activities with high dislikes are basketball, gymnastics, ballet and rugby. One of the reason for a student dislikes rugby is shown in line 212-213. How about basketball, ballet and gymnastics?

A quote was included for rugby on lines 214-216 to demonstrate that a lack of understanding of an activity was one of the most prominent reasons why a child reported to dislike an activity, not for the fact that rugby was one of the most frequently reported activities that children disliked. It is for this reason that we have not included further quotes for basketball, ballet, and gymnastics.

5

Line 214: Figure S3 --> letter e was forgotten

The spelling of ‘figure’ has now been corrected.

6

Reference 29: 26th

We have altered the formatting of this reference so that the date is now in superscript.

7

Reference 57: 28th

We have altered the formatting of this reference so that the date is now in superscript.

he attachment.

Reviewer 2 Report

Overall, this is a well written manuscript that was a pleasure to review. These types of co-creation research ventures have many challenges, and I commend the authors for taking this on. I have some suggestions below to enhance the readability, and impact of this important contribution to the literature. 

Line 142-143. I think it would be valuable to add in that the DDDA is a well-regarded approach within design thinking with a reference. Many interested readers may not be well-versed in design thinking approaches and this well help create a quick connection between this methods statement and the objective.

Table 1 in middle section of the first data row, the fourth bullet says "High fiving or clapping a classmate", I think a word is missing in there (clapping for a classmate, perhaps?)

In Table 1, there are several words on multiple lines cre-ation and stabil-ity for example are split onto two lines. This may be a function of the table formatting, but it would be more readable if the words were on one line

For the supplementary materials - a few of the figures appear to be in landscape format on portrait format pages.

Lines 172-178, while I understand the intent to follow the co-creation process, it is really unclear how the data were clustered, who completed that process (was it co-creation or was it researcher driven?), and how decisions were made. More detail is needed to clarify how this process took place, as there is a significant amount of data that has been organized in various ways, but only a statement that the data were categorized into broad feedback areas as the "how" this data was organized.

Lines 253-261: The conclusions within this statement appear contrary to the way the results are presented in Table 3. The text indicates that parent/caregiver support led to [PE] being perceived as not cool, and influencing body image, but these two outcomes are categorized under children's social influences. 

Page 15 uses the language "group 1's idea" and page 16 uses the language "idea 1" - parity across sections is needed.

More detail into how the researchers approached inputting a third idea in would be helpful. As the intent was to stay true to the co-creation process, this step outside of that process should be well detailed. This is particularly pertinent given the statement on line 422 that stakeholders felt a lack of ownership over the project.

The conclusion could be bolstered by including key take home messages for other researchers, rather than a general "we provided recommendations for researchers", can you summarize the main findings for the reader?

Author Response

Responses to reviewers’ comments

Reviewer 2

Comment #

Comment

Response

1

Overall, this is a well written manuscript that was a pleasure to review. These types of co-creation research ventures have many challenges, and I commend the authors for taking this on. I have some suggestions below to enhance the readability, and impact of this important contribution to the literature.

Thank you for these positive comments.

2

Line 142-143. I think it would be valuable to add in that the DDDA is a well-regarded approach within design thinking with a reference. Many interested readers may not be well-versed in design thinking approaches and this well help create a quick connection between this methods statement and the objective.

Thank you for this suggestion, we agree that this would be helpful for readers who are not familiar with design thinking approaches. Lines 148-149 now read “The DDDA is a well-regarded approach within design thinking and has been used in other school-based co-creation research [42]”.

3

Table 1 in middle section of the first data row, the fourth bullet says "High fiving or clapping a classmate", I think a word is missing in there (clapping for a classmate, perhaps?)

We have changed this sentence, this now reads “High fiving or clapping for a classmate”.

4

In Table 1, there are several words on multiple lines cre-ation and stabil-ity for example are split onto two lines. This may be a function of the table formatting, but it would be more readable if the words were on one line

This is indeed a function of the journal template. To address this comment, automatic hyphenation has been removed in the tables throughout the manuscript to enhance their readability.

5

For the supplementary materials - a few of the figures appear to be in landscape format on portrait format pages.

The supplementary material file has been converted to pdf format and has been re-uploaded so that the content does not move. Thank you for letting us know.

6

Lines 172-178, while I understand the intent to follow the co-creation process, it is really unclear how the data were clustered, who completed that process (was it co-creation or was it researcher driven?), and how decisions were made. More detail is needed to clarify how this process took place, as there is a significant amount of data that has been organized in various ways, but only a statement that the data were categorized into broad feedback areas as the "how" this data was organized.

We would like to clarify that this section is about the interview questions and not any data, and we agree that the section could be made clearer. The interview questions were developed by the research team based on what was observed in the workshops and so for this reason the questions were researcher driven. Our observations highlighted that the main areas of focus for the interview questions should relate to stakeholder feedback, engagement, and enjoyment of the workshops. Lines 175-178 now read, “The interview questions were developed by the research team following their observations of the workshops. These questions were reflective of the stakeholders’ feedback, engagement, and enjoyment of the workshops (see interview guide in Figure S4)”.

7

Lines 253-261: The conclusions within this statement appear contrary to the way the results are presented in Table 3. The text indicates that parent/caregiver support led to [PE] being perceived as not cool, and influencing body image, but these two outcomes are categorized under children's social influences.

Thank you for highlighting this. We agree that the statement appears contrary to the way the results are presented in Table 3. We have edited Table 3 by moving ‘uncool’ from the ‘Children’s social influences’ row to the ‘Lack of parent/carer support’ row to support the statement in lines 261-265.

8

Page 15 uses the language "group 1's idea" and page 16 uses the language "idea 1" - parity across sections is needed.

We have altered page 16 so that there is consistency across sections. Lines 305-306 now read, “Group one’s idea received 1 vote, group two’s idea received 4 votes and the hybrid idea 3, which had some researcher input, received 6 votes”.

9

More detail into how the researchers approached inputting a third idea in would be helpful. As the intent was to stay true to the co-creation process, this step outside of that process should be well detailed. This is particularly pertinent given the statement on line 422 that stakeholders felt a lack of ownership over the project.

We agree with this suggestion that more detail is required into how the researchers approached inputting a third idea. We have provided further information on this in lines 296-309 which now reads, “Because the groups’ ideas were abstract with a limited focus on FMS, the research team were unable to translate them into a deliverable programme. Further, the research evidence presented to the stakeholders was generally not applied, and the research team felt that this would limit the likelihood of intervention feasibility and subsequent effectiveness. Therefore, the research team generated a third programme idea for stakeholders to consider. The ‘hybrid’ programme combined ideas from both groups and included elements reflecting the research evidence presented in workshop 3, to ensure that the instruction and teaching of FMS was embedded alongside pedagogical principles. Due to the time of year (summer term), it was not possible to extend workshop 3 or to schedule a further workshop to engage with the stakeholders to get their input into the hybrid idea 3 before the end of term. The hybrid idea 3 was communicated to stakeholders in the online pre-recorded presentation in workshop 4”. We discuss this decision and its implications in the Discussion section of the manuscript.

10

The conclusion could be bolstered by including key take home messages for other researchers, rather than a general "we provided recommendations for researchers", can you summarize the main findings for the reader?

The take home messages are stated at the end of each discussion paragraph, following signposting at the beginning of the discussion. To set the reader up for this, we have changed lines 330-332 which reads, “These are discussed below and based on the lessons learned from our experiences, suggested recommendations for researchers planning to follow a similar process are presented at the end of each section”.

To avoid repetition in the conclusion, we have briefly stated the main take home messages rather than providing a general summary.

Lines 520-526 now read, “However, there may be certain characteristics of school-based co-creation projects that share some commonalities, which we reflect in our recommendations for other researchers. These relate to investing sufficient time at the start of the process to engage with stakeholders to clarify the co-creation process and its aims, gaining context-specific understanding of the time and resources needed, understanding and managing stakeholders’ roles, commitments, and expectations (including those of the research team), and overcoming any early conflicts. Engaging with stakeholders at all stages of co-creation projects in an open and clear way is crucial to shared success”.

Additional References

[42] Daly-Smith, A.; Quarmby, T.; Archbold, V.S.; Corrigan, N.; Wilson, D.; Resaland, G.K.; Bartholomew, J.B.; Singh, A.; Tjomsland, H.E.; Sherar, L.B.; Chalkley, A. Using a multi-stakeholder experience-based design process to co-develop the Creating Active Schools Framework. Int. J. Behav. Nutr. Phys. Act. 2020, 17, 1-12.

Round 2

Reviewer 2 Report

The revised version of the paper is excellent.